# Liveweight and Sex Effects on Instrumental Meat Quality of Rubia de El Molar Autochthonous Ovine Breed

**DOI:** 10.3390/ani11051323

**Published:** 2021-05-05

**Authors:** Eugenio Miguel, Belén Blázquez, Felipe Ruiz de Huidobro

**Affiliations:** Departamento de Investigación Agroalimentaria, Instituto Madrileño de Investigación y Desarrollo Rural Agrario y Alimentario (IMIDRA), Finca el Encín. Apartado 127, Autovía A-2, Km 38,200, 28800 Madrid, Spain; blazquezcmb@madrid.es (B.B.); felipe.ruiz@madrid.org (F.R.d.H.)

**Keywords:** lamb meat, weight, sex

## Abstract

**Simple Summary:**

A total of 84% of the autochthonous livestock breeds in Spain are in danger of extinction. This decline is mainly produced because they are not very productive and the farmers have been oriented to the most productive breeds, neglecting the rest. There are many factors that push us to conserve autochthonous breeds with little census. These animals are very adapted to their environment, so they are the ones that best exploit the geographical characteristics of their area; besides, their products have a high quality and these breeds have very beneficial genetic components: they are a source of very useful and not sufficiently known gene variants that make them a genetic heritage of undoubted value. If they become extinct, we would be losing an irreplaceable genetic quality. Rubia de El Molar sheep, an autochthonous breed of Madrid, Spain, is currently in serious danger of disappearing. In this work, the meat quality of lambs from Rubia de El Molar breed was studied to test both weight and sex effects on meat quality, assessed by means of instrumental methods. This is the first work that studies the instrumental quality of Rubia de El Molar suckling lamb meat.

**Abstract:**

The effects of sex and weight on instrumental meat quality characteristics of Rubia de El Molar autochthonous ovine breed were studied. Four weight groups (10, 15, 20 and 25 kg, each of the groups with seven males and seven females) were assessed. A decrease of longissimus thoracis muscle (LT) lightness from 43.39 for 10 kg lambs to 38.84 for 25 kg group was observed. LT redness and yellowness indices and chromaticity values increased as weight increased. The percentage of juice expelled increased from 11.52 (10 kg) to 17.15 (15 kg). The percentage of intramuscular fat in biceps femoris (BF) and supraespinatus (SE) muscles increased as weight increased. Warner–Bratzler shear force (WBSF) in raw meat and hardness in cooked meat increased as liveweight increased. However, a decrease in the amount of soluble collagen in quadriceps femoris (QF), biceps femoris (BF) and infraespinatus (IE) muscles was observed as weight increased. Sex effect was only observed for intramuscular fat content in QF and BF muscles (2.4% in males and 2.9 in females) and WBSF in raw meat (26.12 N for males and 21.25 N for females). Weight had a greater effect than sex on Rubia de El Molar meat quality characteristics.

## 1. Introduction

Rubia de El Molar sheep, an autochthonous breed of Madrid (Spain), is currently in serious danger of disappearing [1]. Local indigenous breeds (like Rubia de El Molar) have suffered years of neglect, which has led in some cases to their disappearance, with a loss of not only a gross genetic potential, but also a part of our history. The big problem was its low productivity. This situation resulted in a systematic and indiscriminate crossbreeding with animals of other, more producing breeds. There are currently only ten herds that raise animals of this ovine breed in Madrid and also in the world [1]. In addition to Imidra’s herd, only one other herd exclusively raises animals from the Rubia de el Molar breed. The others raise animals of Rubia de El Molar breed and also animals from Assaf, Lacaune, Manchega or Colmenareña ovine breeds. Only one of these flocks milks its animals and uses the milk obtained for the production of sheep’s yoghourt. The rest are sheep meat farms. Rubia de El Molar is a dual-purpose (milk/meat) sheep [1]. At present, meat production accounts for a large proportion of the economic value of this breed of sheep. Lambs are almost entirely used for consumption as suckling (young lambs fed exclusively on maternal milk). There is a great demand for Rubia de El Molar meat by restaurateurs.

Different factors are known to influence lamb meat quality, such as slaughter weight, breed, sex, diet composition, genetic factors, feeding system, age of weaning and individual cues [2,3,4,5]. Carcass weight is an important cue in the categorization of lamb meat. In the Mediterranean countries, lambs are slaughtered at very young ages producing light carcasses. These animals receive concentrate feed in addition to their mother’s milk. In Spain, the market demands light carcasses, under 11 kg [2], and meat from light lambs is considered better (more tender and less intense flavor) than meat from heavier animals. Sex is also one of the most important factors influencing carcass and meat quality. Effects of sex and genotype trend to disappear as growth and development increase, but when animals are slaughtered at the same live weight, sex is an important factor. Female lambs produce more subcutaneous fat and show a higher degree of fatness than males [6]. At the same live weight, females have achieved a higher percentage of adult weight, a higher development stage and a higher degree of fatness. Weight [7] and sex [8] have an effect on the prediction of tissue composition in suckling lamb carcasses according to the European Union scale.

In the present work, the meat quality of lambs from Rubia de El Molar breed is studied to test both weight and sex effects on meat quality, assessed by means of instrumental methods. Meat quality is studied by different physical-chemical parameters, pH and pH drop, color, water holding capacity (WHC), cooking loss (CL), moisture content (MC), intramuscular fact content (IMF), collagen content (total and soluble collagen; TC and SC) and texture instrumental parameters, both in raw and in cooked meat, by means of two tests: Warner–Bratzler shear force test (WBSF) and texture profile analysis (TPA). This is the first work that studies the instrumental quality of Rubia de El Molar meat, currently in serious danger of disappearing.

## 2. Materials and Methods

### 2.1. Animals

Fifty-six Rubia de El Molar breed lambs were used in this work. All the animals studied in this work came from an experimental herd of lamb of Rubia de El Molar ovine breed, conserved in “El Encín” farm belonging to the Madrid Institute of Agricultural and Food Research (IMIDRA). The animals were reared in an intensive regime. Studied treatments have been sex (males and females) and slaughter live weight (10, 15, 20 and 25 kg of live weight). Animals were milk fed until slaughter time. A compound feed was supplied to the lambs ad libitum, in addition to ewe’s milk. Most of the lambs selected for this study came from single birth, except for 9 of them that were twins but were raised as single lambs. Animals were slaughtered in a commercial slaughterhouse, at live weight intervals of 5 kg (from 10 to 25 kg) in accordance with the rules of Spanish legislation regarding the transport and slaughter of meat animals [9]. Seven males and seven females were included in each slaughter weight group.

### 2.2. Sampling

At 24 h from slaughter, after a chilling at 4.0 ± 3.0 °C in a chilling chamber, carcasses were split and left halves were jointed. During the dissection process, for the subsequent physicochemical analysis of the meat, a series of muscles were selected from each of the dissected joints. Some muscles were extracted and vacuum packed for freezing at −40 ± 2.0 °C, temperature at which they were kept until the moment of the corresponding analysis. Others were kept at 4.0 ± 3.0 °C in a chilling chamber for in fresh assays. Frozen samples were thawed (before two months after slaughter) and analyses were performed.

Portions of the muscle longissimus thoracis et lumborum (LTL) were taken as follows: LTL, from its beginning to the level of the fifth thoracic vertebra, was taken for conducting WHC and moisture assays. These techniques were performed in fresh meat, 24 h after slaughter time. LTL between the sixth and twelfth thoracic vertebrae was used to perform texture instrumental analysis. Muscle belonging to the right ribs was used for texture analysis in cooked meat, while the muscle belonging to the left side ribs was used for texture analysis in raw meat. Meat color was measured at the level of the thirteenth thoracic vertebra (right side).

Three muscles were selected from the legs: quadriceps femoris (QF), biceps femoris (BF) and semimembranosus (SM). These three muscles were used for the chemical extraction of intramuscular fat (IMF) and for the analysis of total (TC) and soluble (SC) collagen content. In the shoulder, two muscles were selected: supraespinatus (SE) (for chemical extraction of IMF) and infraespinatus (IE) (for TC and SC content analysis). The reason for choosing these muscles is that the lamb is so small that it is consumed as whole joints and joints always include several muscles. Joints have been chosen from three different categories (extra, first and second) and within each one, the most convenient muscles to work with. According to muscle-fiber type, longissimus dorsi, biceps femoris and semimembranosus are fast-twitch red muscles, having a higher ATPase activity and both high oxidative and high glycolytic activity. Supraspinatus and infraspinatus are essentially oxidative slow-twitch muscles having low ATPase activity. Finally, quadriceps femoris and semitendionosus are essentially glycolytic fast-twitch muscle, having the highest ATPase activity.

### 2.3. pH

A Crison pH meter equipped with a penetration probe was used. Measurements were taken on semitendinosus (ST) and longissimus thoracis muscles (in the last rib) at 45 min post-mortem (just after carcass dressing) and after a 24 h chill, watching the pH evolution (pH drop). Measurements were performed in the muscle, into a scalpel cut on its surface. Adjustment of the pH electrode was performed immediately before use. Two Crisson™ standard solutions (“calibrators”) were used (covering the required working range of pH values): pH = 7.00 and pH = 4.01, at 25 °C, so pH readings were made by interpolation and not by extrapolation.

### 2.4. Color

Meat samples were kept at 4 °C for daily color measurement for 5 days post-mortem. After removing the LTL muscle from the carcass and having it cut at the thirteenth thoracic vertebra, color measurements were performed on the surface of the section, after waiting one minute to let fresh air in contact with the meat. A colorimeter Minolta Chroma Meter CR-200 is used, using the trichromatic coordinates of CIELAB color space [10] (CIE, 1986) with the 10° Supplementary Standard Observer, the D65 Standard Illuminant and an 8 mm diameter measuring area. This system allows to identify color with the aid of the coordinates L* (lightness), a* (redness) and b* (yellowness). From these coordinates, the colorimetric indices are obtained:Chromaticity (C*) = (a*^2^ + b*^2^)1/2, (1)
and
Hue (H*) = arctan (b*/a*) ∗ 57.29). (2)

### 2.5. Moisture, Water Holding Capacity (WHC) and Cooking Loss (CL)

The moisture determination was performed according to International Standard ISO R-1442 (ISO, 1973) [11] and the Spanish Methods for Analysis of Meat Products (BOE, 1979) [12]. WHC was assessed according to [13], who measured the water expelled by subjecting the meat to an external force. Results were expressed in percentage of water expelled. CL determines the fluid released upon heating of the meat, without applying external forces. LT muscle samples were introduced into a polyethylene bag unlocked, placing it in a water bath at 90 °C. In each sample, the core temperature in each piece was measured by a thermocouple, the pieces being removed from the bath once the water reached 75 °C. Samples were removed from the bags, slightly dried with filter paper (without pressing at all) and weighed. Results are expressed as a percentage of sample weight loss from the initial sample weight.

### 2.6. Collagen Content

Total collagen (TC) measurement was based on the colorimetric determination of hydroxyproline (Hyp). Samples of QF, BF, SM and IE muscles, were thawed and trimmed of fat, fascia and visible connective tissue and were used for TC determination. Meat proteins were hydrolyzed in an acid medium (sulfuric acid) and heated, so that residues of hydroxyproline which are released are oxidized by the action of chloramine T. Pyrrole derivatives are generated and, after addition of p dimethylaminobenzaldehyde, they resulted in a colored compound. Soluble collagen (SC) determination was based on the method of Hill [14]. The ground of this method involves the quantification of the total hydroxyproline and of the hydroxyproline extracted after performing a heat treatment of the sample. Difference in value of total and SC may be obtained, the subtraction giving the amount of insoluble collagen (IC). To determine the SC, samples of QF, BF and IE muscles were chopped and freed from all connective tissue. Fifteen milliliters of distilled water were added and the mixture was shaken for 2 h in a water bath at 77 °C. Collagen soluble and insoluble fractions were separated by centrifugation. The SC was expressed in relation to the total collagen content of the total sample collagen: %SC = SC/(IC + SC).

### 2.7. Intramuscular Fat Content (IMF)

Extraction of fat was performed according to [15]. Lipid extraction of IMF was carried out on samples obtained from the muscles QF, BF, SM and SE, as stated above.

### 2.8. Instrumental Texture

Vacuum-packed and frozen (−40 °C) loins were cut into 1.5 cm-thick slices, at right angles to the longitudinal axis of the loin. Slices were kept frozen under the same conditions until the day of analysis. On this day, samples were thawed by submersion in cold water and then a series of 1 × 1 cm-section parallelepipeds were obtained (for analysis of texture in raw meat). For texture analysis in cooked meat, other slices (once thawed) were wrapped in aluminums foil and cooked in a 2 side-grill preheated to 300 °C until they reach an internal temperature of 75 °C. Once cooked, 1 × 1 cm^2^-section parallelepipeds were prepared. Five or 6 readings were made in the texturometer and the average was the result of each muscle of each lamb.

To perform texture analysis, two types of tests were executed: a shear force test, using a Warner–Bratzler probe (WBSF) and a compression test in a texture profile analysis (TPA) with a cylindrical ebonite probe of 10 mm diameter, assessing hardness, springiness and chewiness. A texture analyzer TA-XT2™, with the software Texture Expert™ release 1.19 for Windows (Stable Micro Systems, Surrey, UK), was used to perform this test.

### 2.9. Statistical Analysis

Statistical analysis was performed by a one-way ANOVA with Statistica software package release 5.0 [16,17]. The effect of sex and weight on the instrumental quality of the Rubia lamb meat from El Molar was studied. An analysis of variance was carried out considering sex and weight as fixed effects. The interaction of both factors was also studied. In the case of the effect of weight, after the analysis of variance, a Tukey test for multiple comparisons was performed.

To study the effect of the aging time on the colorimetric parameters of Rubia de El Molar lamb meat, an analysis of variance was carried out considering ageing time as a fixed effect. After the ANOVA, a Tukey test for multiple comparisons was carried out to detect differences between the groups of animals studied. Sex was taken into account and 4 groups of animals were selected according to weight: 10, 15, 20 and 25 kg, regardless of whether they were male or female and other two groups according to sex: male and female groups, regardless of whether they weighed 10, 15, 20 or 25 kg. An analysis of variance was performed for each of these groups, considering ageing time as a fixed effect.

## 3. Results

### 3.1. pH

Weight (Table 1) influences the pH of the LT muscle of Rubia de El Molar ovine breed after 24 h of cooling at 4 °C. Lambs slaughtered at 20 kg showed a higher pH than the rest of animals (*p* ≤ 0.01). However, the pH drop was higher in lambs slaughtered at 15 kg (*p* ≤ 0.05).

Sex had no effect on LT muscle pH of Rubia de El Molar ovine breed (measured both 45 min and 24 h post-mortem) and on pH drop (Table 1).

Table 1 also shows the mean and the variance analysis of the Rubia de El Molar ST muscle pH values obtained measured at 45 min and 24 h post-mortem and the pH drop, in function of slaughter liveweight and sex. Slaughter liveweight only showed a light effect on the pH values obtained at 24 h post-mortem, so the meat of lambs being slaughtered at 20 kg had a higher pH (*p* ≤ 0.05) than the meat of the rest of the animals studied. No effect of sex on ST muscle pH or on pH drop was observed.

### 3.2. Color

Table 2 shows the mean and variance analysis of the LT of Rubia de El Molar colorimetric parameters measured during the 5 days after slaughter, as a function of slaughter liveweight and sex. Slaughter liveweight had a significant effect (*p* ≤ 0.01) on the L* of LT muscle every day. As weight increased, a decrease of L* was observed; thus, the lambs slaughtered at 10 kg showed a brighter meat and the lambs slaughtered at 25 kg were those with darker meat. Slaughter liveweight had a significant effect (*p* ≤ 0.01) on a* during the three first days of ageing. After 24 h post-mortem, LT a* increased as weight increased (12.81, 13.84 and 15.45 for 10, 15 and 20 kg groups, respectively). After 48 h post-mortem, an increase of a* is only observed for 15 and 20 kg groups (from 12.48 to 14.09 at two days of ageing and from 12.69 to 14.25 at three days of ageing). After four days of ageing, no statistically significant differences between weight groups were observed.

Slaughter weight only had a significant effect on b* at 48 h post-mortem (*p* ≤ 0.05), increasing its value from 4.92 (10 kg) to 6.17 (15 kg), 6.25 (20 kg) or 6.10 (25 kg). No statistically significant differences were measured between the 15, 20 and 25 kg groups.

However, sex did not have an effect on LT muscle L*, a* or b* parameters.

An increase in C* values between 10 kg and 20 kg was detected, both at 24 and 48 h of ageing. At 72 h and 96 h post-mortem, the increase occurred between 15 kg and 20 kg liveweight at slaughter (*p* ≤ 0.001). At the fifth day of ageing, statistically significant differences between lambs slaughtered at 10 kg and 20 kg were observed. Sex did not an effect on LT C*. H* values for LT muscle were not affected by slaughter liveweight or sex during the five first days of ageing.

Table 2 shows the mean and the variance analysis of the colorimetric parameters in longissimus dorsi muscle as a function of ageing time.

For L*, variation was only observed for animals slaughtered at 15 kg live weight. L* increases between 24 and 48 h post-mortem in these lambs (*p* ≤ 0.01), while from 72 h of maturation, the colorimetric values of L* do not differ statistically from those obtained at 24 and 48 h post-mortem. No effect of sex on L* variation with ageing was detected. No significant changes were observed in the L* parameter of the lamb meat during the first 5 days of maturation, neither in the case of males or females.

Redness index decreased in lambs slaughtered at 20 kg and 25 kg live weight between 24 h and 48 h post-mortem (*p* ≤ 0.01), from a value of 15.45 for lambs slaughtered at 20 kg at 24 h post-mortem to a value of 14.09 at 48 h post-mortem, and from 15.67 in lambs slaughtered at 25 kg live weight at 24 h post-mortem to a value of 14.16 at 48 h. For 10 and 15 kg groups, no statistically significant differences in a* in function of time of ageing were observed. Redness index also decreased in male lambs between 24 h and 48 h post-mortem. No differences were observed between 2 and 5 days of ageing. For female group, a similar trend was detected but the differences were not statistically significant. At 3 days of ageing, an interaction sex x weight was detected. There were no differences in the red index between males and females for the weights of 10, 15 or 20 kg, but there are for the animals weighing 25 kg. The *p*-values for the effect of sex decreased as the weight of the animals increased (0.898, 0.545, 0.233 and 0.013 for 10, 15, 20 and 25 kg groups, respectively). For the 25 kg group at 3 days of ageing, the a* value was 13.22 for male lambs and 15.21 for female lambs.

For b*, statistically significant differences for all weight groups studied were detected (*p* ≤ 0.001). The greatest differences were obtained between 24 h and 48 h, resulting in an increased value of the b* during ageing. No statistically significant differences were shown for C* as a function of ageing time.

The time elapsed after slaughtering had a significant effect on the mean values of H* parameter in all weight groups (*p* ≤ 0.001). These values increased significantly between 24 h and 48 h post-mortem in all weights studied and between 24 h and 72 h post-mortem in lambs slaughtered at 20 kg.

### 3.3. Moisture, Water Holding Capacity and Cooking Loses

Table 3 shows mean values for moisture (expressed as a percentage), WHC (expressed as percentage of water expelled) and CL (expressed as a percentage) and the variance analysis performed as a function of slaughter liveweight and sex for Rubia de El Molar ovine breed. No effect of sex on moisture, WHC or CL was detected. In addition, no effect of slaughter weight on MC was observed (Table 3).

WHC, expressed as a percentage of juice expelled under pressure, was affected by slaughter liveweight; it was noted that 20 kg and 25 kg lambs had a higher proportion of liquid expelled (15.39% and 17.15% respectively) and, therefore, were less able to retain water (*p* ≤ 0.001) than 10 kg and 15 kg lambs (11, 52% and 13.15%, respectively). CL was greater for the 15 kg lamb group than for the rest of the animals. There was an increase in CL when cooking meat when the weight increased from 10 to 15 kg and a decrease in cooking losses when the weight increased to 20 and 25 kg

Rubia de El Molar meat WHC was not affected by sex (Table 3). Finally, slaughter weight influenced CL so that 15 kg lambs showed the greatest loss (*p* ≤ 0.01), differentiating these animals from the rest. Sex did not have any effect on CL.

### 3.4. Intramuscular Fat

Table 4 shows mean values for the percentage of IMF samples obtained from the QF, BF, SM and SE muscles and the variance analysis performed as a function of slaughter liveweight and sex for Rubia de El Molar ovine breed. The percentage of IMF was affected by slaughter liveweight in BF (*p* ≤ 0.001) and SE (*p* ≤ 0.001), resulting in a significant increase of IMF at 15 kg and 20 kg (from 2.31% to 2.97% at 15 kg and 20 kg, respectively, in BF and from 2.61% to 3.49%, respectively, in SE muscle) (*p* ≤ 0.001).

Rubia de El Molar ovine breed females had a higher proportion of IMF in the QF and BF muscles (*p* ≤ 0.05) than males, with no differences between sexes in IMF for SM and SE muscles (Table 4).

### 3.5. Collagen

The percentages of TC, IC and SC extracted from the QF, BF and IE muscles and the analysis of the variance in function of slaughter live weight and sex are reflected in Table 5.

In QF muscle, lambs slaughtered at 10 kg had a higher proportion of TC (0.68%) than those slaughtered at 20 kg (0.52%) (*p* ≤ 0.01). For IE muscle, 10 kg lambs had the highest proportion of TC (1.03% vs. 0.79%, 0.64% and 0.65% observed for 15 kg, 20 kg and 25 kg animals, respectively) (*p* ≤ 0.01). Sex only influenced the proportion of TC level of the QF muscle with the males who had the highest percentage of TC (0.66% versus 0.54% of females) (*p* ≤ 0.001).

IC content was affected by slaughter liveweight for BF muscle, although the differences were minimal (*p* ≤ 0.05) and were only observed in animals slaughtered at 15 kg and 25 kg (0.53% and 0.67% for those living slaughter weights, respectively). Sex had an effect on the percentage of IC in BF, with the males having the highest percentage of TC (0.63% versus 0.54% of females) (*p* ≤ 0.05).

Weight showed a great influence on the percentage of SC in all muscles studied (*p* ≤ 0.001). In QF muscle, the amount of SC decreased from 0.27% in 10 kg group to 0.18% in animals slaughtered at 20 kg. In BF and IE muscles, there was also a decrease in the amount of SC as weight increased. A statistically significant decrease was observed between 15 kg and 20 kg groups (from 0.28% to 0.22% for BF and from 0.31% to 0.23% for IE muscle, respectively). It is generally considered that, as the carcass weight increases, the percentage of soluble collagen decreases and that is one of the reasons why WBSF increases as the carcass weight increases. Therefore, heavier lambs can produce less tender meat because solubility of intramuscular collagen decreases parallel to weight.

Our results showed that the SC values obtained in the three muscles studied in the Rubia de El Molar meat were similar both in males and females and no statistically significant differences were found between sexes.

### 3.6. Instrumental Texture

The mean values of instrumental texture parameters both in raw and in cooked meat of Rubia de El Molar ovine breed are shown in Table 6, as well as the analysis of variance of the instrumental texture parameters in the function of slaughter live weight and sex.

#### 3.6.1. Raw Meat

Slaughter weight had a significant effect only on WBSF in raw meat (*p* ≤ 0.05). Differences arose between 10 kg and 20 kg groups. Higher values were obtained for 25 kg animals. Males were also generally those who had the highest values of WBSF (*p* ≤ 0.001).

The results of the parameters hardness, springiness and chewiness in texture profile analysis (TPA) were not affected by either weight or sex.

In WBSF in raw meat the interaction W **×** S resulted significant. Differences due to sex were observed for the groups of lambs of 20 and 25 kg, but not for those of the weights of 10 and 15 kg. Sex effect on WBSF was greater as weight increased. *p*-values for sex effect were 0.943 (10 kg), 0.443 (15 kg), 0.001 (20 kg) and 0.067 (25 kg). For light lambs (10 and 15 kg in weight) there were no differences in WBSF between the groups of male and female lambs. However, for the heaviest lambs, an effect of sex could be appreciated, so that the meat of male lambs is tougher than that of females. The values of the WBSF for the meat of male lambs of 20 kg was 31 N, whereas for the meat of female lambs, it was 20 N. For the meat of female lambs of 25 kg was 29 N, whereas for the meat of female lambs was 22 N.

#### 3.6.2. Cooked Meat

No effect of weight or sex on the results of Braztler–Warner Shear Force test breaking force was observed. When studying cooked meat texture properties by means of TPA test, only a small effect of weight on hardness was obtained. A statistically significant increase in hardness is observed from 10 kg to 25 kg (*p* ≤ 0.05). Sex had no effect on instrumental texture parameters of cooked meat.

## 4. Discussion

Live weight influences the pH of Rubia de El Molar LT muscle after 24 h of chilling at 4 °C. Lambs slaughtered at 20 kg showed a higher pH than the rest of animals.

The effect of carcass weight on lamb meat pH has been widely studied with diverse results. As carcass weight is increased, a decrease in pH at the time of slaughter and 45 min later was measured in Lacha breed lambs, with correlation coefficients between 0.27 and 0.59, depending on the muscle [18]. Significant differences were also found when comparing pH values from meat of different slaughter weights of two Canarian sheep local breeds (Canarian breed and Canarian-Hair breed) of 10, 16 and 25 kg [19]. The lambs of 10 kg showed the higher pH values [19]. Conversely, Sañudo et al., (1996) found in Rasa Aragonesa breed lambs that, as carcass weight increased (from 8 to 13 kg), an increase in the final meat pH value (5.58 vs. 5.86) was observed [20]. Heavier lambs (Mirandesa and Bragançana lambs) had also a significant higher pH at 24 h post-mortem than lighter lambs [5]. However, Sañudo et al., (1993) [21] and Vergara et al., (1999) did not find an effect of weight on meat pH [22].

Sex had no effect on LT muscle pH of Rubia de El Molar ovine breed (measured both 45 min and 24 h post-mortem) and on pH drop (Table 1). These results are consistent with those previously obtained [20] that showed a little effect of sex on meat pH in sheep. However, in suckling lambs of Lacha breed, López (1987) found that females’ final pH of LT muscle tended to be higher than males’ final pH [18]. For heavier lambs, no significant differences in pH values due to sex were observed. Moreover, Dransfield et al., (1990) found significant differences in final meat pH values between ram lambs, wether lambs and ewe lambs in Suffolk breed [23].

A weight effect on Rubia de El Molar meat color was observed. As weight increased, a decrease of L* and an increase of redness index is observed. No effect of sex was observed. A similar effect was observed in Mirandesa and Bragançana lambs [5]: meat L* decreased with increasing live weight and the light lambs had higher b* than the heavier ones. As slaughter weight increased, meat L* decreased also for two Canary local sheep breeds, Canarian breed (CB) and Canarian-Hair breed [19]. Sañudo et al., (1996) found that estimates of L* of LT muscle of Rasa Aragonesa lambs showed no significant differences between 8 kg and 10 kg groups (carcass weight), which were significantly different from 13 kg, which was darker [20]. For a*, a similar pattern was observed: carcasses from 8 kg had significantly lower values than those from 10 and 13 kg groups.

Older animals (slaughtered at 60 days of age and 12.44 kg) had meat with higher a* value (19.43 vs. 18.91) versus meat from younger animals (40 days and 10.36 kg) in an assay of Bergamasca lambs reared according to the traditional transhumant system in central Italy [24]. Beriain et al., (2000) explained that for Rasa Aragonesa and Lacha breeds, meat was redder at 20 kg than 10 kg [25]. However, Martínez-Cerezo et al., (2005) showed that the increase from 20 to 30 kg did not affect meat redness [26].

A decrease in L* associated with an increase in weight in samples of Manchega breed was observed. An increase in a* as weight increased was measured, but differences failed to reach statistical significance [22]. Meat from light and moderate carcass weight (10–15 kg) lambs had higher lightness values than meat from heavy carcass weight (20–25 kg), while redness was greater for heavy carcass weight and female lambs, compared with light and moderate carcass weight and male lamb groups [27].

Meat of high lambs (35–36 kg) had lower L* value than medium (30–31 kg) and low weight (25–26 kg) in an assay with fifty-two male Kivircik lambs to investigate the influences of weight and production system (concentrate-based system, or pasture-based system) on carcass and meat quality characteristics [28].

In general, b* parameter was not affected by weight for Rubia de El Molar lamb meat. In a study of Rasa Aragonesa, higher b* values were found in light lambs (20–22 kg slaughter weight) than in heavy lambs (30–32 kg) [26].

Color intensity increases with weight because of the increase in myoglobin concentration [20]. This increase is rapid in the early stages of animal development, then stabilizes. López (1987), observed an increased muscle myoglobin concentration for 12 and 24 kg liveweight for Menhaden breed lambs [18].

Sañudo et al., (1993) found in Rasa Aragonesa, Merino and Lacaune breeds a greater color intensity, amount of pigment and a* and lower L* in heavy weight lambs (28–30 kg BW) than in light lambs (23–25 kg BW) [21]. The increase in carcass weight (7.5 to 15.5 kg) carries a darker meat, which is reflected in the higher concentrations of myoglobin for heavier lambs; in parallel, it is observed that the relationship b*/a* decreases with age, so changing the color characteristics of meat leads to darker shades of pink [20,22].

Sex did not have an effect on LT muscle L*, a* or b* parameters. Ekiz et al., (2019a) in an assay to determine the effects of sex on carcass and meat quality of Kivircik and Karacabey Merino lambs published that differences between male and female lambs for b* and hue angle values were not significant [29]. However, LT samples of female lambs had lower L* and greater a* and chroma values than male lambs. L* values were influenced by sex, with entire males presenting higher values than castrated males and females in an experiment conducted to evaluate the effects of feed restriction (FR) and sex on the quantitative and qualitative carcass traits of Morada Nova [30].

In some studies, carcass weight did not have a significant effect on any of the color parameters of lamb meat [31,32,33].

As weight increase an increased loss of juice is observed for Rubia de el Molar lamb meat. These results are consistent with other previous works that suggest that increasing carcass weight decreases the WHC (expressed as a percentage of water retained) [34]. Therefore, Sañudo et al., (1996) [20] observed that with increasing live weight, meat has also greater ability to release water. López [18] concluded that increased carcass weight is associated with an increased loss of juice (explaining carcass weight a 53% of the variations observed in the WHC) [18]. CL were greater for the 15 kg group than for the rest of the animals. There was an increase in CL when cooking meat when the weight increased from 10 to 15 kg and a decrease in cooking losses when the weight increased to 20 and 25 kg. This biphasic behavior could be due to a lower water retention capacity (higher percentage of water expelled) for the 20 and 25 kg groups, a higher intramuscular fat content (for the 20 and 25 kg groups) and a greater drop in pH between 45 min and 24 h post-mortem for 15 kg group. This can be also related to collagen development, as collagen solubility decreases as the animal ages. In QF, BF and IE muscles there was a decrease in the amount of SC as weight increased.

Rubia de El Molar meat WHC was not affected by sex. Similar results were obtained by López [18]. Differences between male and female lambs in terms of expressed juice, cooking loss and WB shear force were also not statistically significant [30]. No effect of sex on moisture, WHC or CL was detected. In addition, no effect of slaughter weight on MC was observed (Table 3). Similar results were obtained by [35] in male lambs of two Southern Spanish breeds, a dairy breed (Grazalema Merino sheep) and a meat breed (Churra Lebrijana sheep) (12 kg versus 20 kg). Drip loss was lower for female and heavy carcass weight group than male and light carcass weight group, respectively [27].

Intramuscular fat content of Rubia de el Molar meat increases as weight increased. These results are in agreement with those previously obtained by other authors. Back fat thickness and intramuscular fat (IMF) content were greater for heavy carcass weight (20–25 kg) than moderate carcass weight (15–20 kg) and light carcass weight (10–15 kg) in fat-tailed Chall lambs [27]. A total of 13 kg carcasses meat were fatter (17.01% shoulder fat) than those of 8 and 10 kg groups (14.3% and 15.76%, respectively) [20]. Martínez-Cerezo et al., (2005) described that fat content increased with slaughter weight, following a linear tendency [26]. Meat from light lambs (20 kg) had a higher IMF percentage than that from suckling lambs (12 kg). The increase in IMF content in heavier lambs has been reported by other authors [25,36].

Rubia de El Molar ovine breed females had a higher proportion of IMF in the QF and BF muscles than males, with no differences between sexes in IMF for SM and SE muscles.

Regarding the effect of sex on the IMF of meat, Pannier et al., (2014) [37] found that ewe lambs had significantly higher IMF levels than male lambs. Similar results were obtained by [27]. Interestingly, Craigie et al., (2012) [38] found females to have higher IMF levels compared to males, though rams were used instead of wethers.

In QF and IE Rubia de El Molar muscles, lambs slaughtered at 10 kg had a higher proportion of TC. In QF, BF and IE muscles there was a decrease in the amount of SC as weight increased. IC content was affected by slaughter liveweight for BF muscle, although the differences were minimal. In addition, male lambs had a higher proportion of TC (QF muscle) and IC (BF muscle) than females. Sex only influenced the proportion of TC level of QF muscle, with the males who had the highest percentage of TC.

Martínez-Cerezo et al., (2005) showed that collagen concentrations were mainly influenced by breed and slaughter live weight had only a slight effect on collagen solubility percentage [26]. Significant interactions were found between both effects. Breed also affected insoluble hydroxyproline content. The overall mean for the three slaughter weights was significantly higher in the Churra breed than the Rasa Aragonesa breed or Spanish Merino breed. As reflected by the total and insoluble hydroxyproline contents, in general, the Spanish Merino lambs had the most soluble collagen. The solubility percentages only decreased clearly with increasing slaughter live weight in the Churra breed. On the contrary, the solubility percentages of the Spanish Merino lambs increased with weight. Camacho et al., (2017) [19] shown than collagen solubility was not affected by slaughter weight; this could explain why weight does not affect WBSF for Canarian breed (CB) and Canarian Hair breed lamb meat. These authors showed that sex did have an effect on collagen solubility of meat and higher values were measured in males.

These differences in collagen solubility could be related to the different productive aptitudes of each breed. Specialized meat breeds could deposit less stable collagen cross links and thus have more SC to facilitate greater muscle growth [26].

However, Santos-Silva et al., (2002) [2] did not find significant differences in collagen contents among breeds or slaughter weights, from Merino Branco or Île de Franc ex Merino Branco crossbred lambs slaughtered at 24 or 30 kg.

Collagen content depends on many factors. Within the same species and breed, the collagen content is influenced by both age and type of muscle. Meat quality is important not only collagen content, but collagen solubility also. Solubility decreases as the animal ages [39] and varies with the breed, muscle and gender [40]. Levels of collagen and the relative amounts of highly cross-linked collagen have long been associated with meat quality and meat tenderness [41].

Slaughter weight had a significant effect on WBSF in raw meat and on TPA hardness in cooked meat. As weight increased, meat was harder. Males had the highest values of Warner–Bratzler shear force in raw meat. Sex had no effect on TPA parameters in cooked meat.

Other researchers showed that texture instrumental parameters were affected by carcass weight [35]. In addition, WBSF increased with liveweight [5]. Light carcass weight lambs and female group had lower shear force compared with heavy carcass weight and male groups [27]. WBSF was also significantly affected by carcass weight group in Rasa Aragonesa lambs [20]. Mean WBSF values were higher for muscle samples from 10 kg carcasses than those from 8 and 13 kg groups.

Cloete et al., (2012) [42] reported a 9% increase in shear force for longissimus muscle from rams compared to ewes. The report of Johnson et al., (2005) [43] also showed that SM from rams had significantly higher WBSF values compared to ewes. Small differences were reported by [44] across a number of genotypes with wethers producing significantly tougher meat than ewe lambs over an age range of 4 to 22 months.

Gender differences are also well defined; at the same age, females have the tenderest meat [45,46,47], especially near sexual maturity [47]. However, according to Dransfield et al., (1990) [23], there is no evidence to say that the meat of boars is harder than that of cryptorchid or partially castrated pigs. In sheep, values of WBSF are generally higher in carcasses of males than females [45], which may be due to different levels of fat. It could be possible that these results were age-dependent, because in young lambs (1 to 3 months), no sex effect is observed [48]. In animals of the same age, greater hardness is often attributed to male meat than female [49].

Live weight had also a great effect on Rubia de El Molar meat sensory hardness assessed by trained panelists, as hardness scores grew greater as live weight increased. Heavier animals showed a springier meat and also received lower scores for pleasantness. These sensory differences may be due to the results obtained, which show an effect of weight on WBSF in raw meat and TPA hardness in cooked meat (data not shown).

Rubia de El Molar heavy carcasses (20 and 25 kg) showed a harder and springier meat than light carcasses (10 and 15 kg). In addition, the 10 and 15 kg groups received higher scores for pleasantness (data not shown). Differences observed in instrumental parameters (pH of the LT muscle, WHC, color, intramuscular fat content and WBSF and TPA assays) could explain the differences observed in sensory parameters.

## 5. Conclusions

 Slaughter weight had a greater effect on Rubia de El Molar meat quality than sex. Lambs slaughtered at 20 kg showed a higher meat pH after 24 h of chilling at 4 °C than the rest of animals. As weight increased, a decrease of L* and an increase of redness index were observed. Weight had no effect on meat moisture content; however, animals slaughtered at 20 and 25 kg expelled a higher proportion of water under pressure. In addition, 15 kg animals showed greater cooking loss values than the rest. The percentage of IMF in BF and SE muscles was affected by weight. Lambs slaughtered at 10 kg had a higher proportion of total collagen content in QF and IE muscles. In QF, BF and IE muscles, there was a decrease in the amount of soluble collagen as weight increased. Insoluble collagen content was higher for BF muscle in animals slaughtered at 15 kg and 25 kg. Slaughter weight had a significant effect on WBSF in raw meat and on TPA hardness in cooked meat. Females’ meat had a higher proportion of intramuscular fat in quadriceps femoris and biceps femoris muscles. In addition, male lambs had a higher proportion of TC (QF muscle) and IC (BF muscle) than females. Males had the highest values of Warner–Bratzler shear force in raw meat. Sex had no effect on TPA parameters in cooked meat. As the preferred cooking method for this kind of meat is roasting (shoulder and limb) or grilling (cutlets), collagen content and soluble collagen proportion are crucial for a better gastronomic experience. Results suggest that lambs of 15 kg liveweight would be more tender. Heavier lambs will have more collagen content and a tougher meat.

## Figures and Tables

**Table 1 animals-11-01323-t001:** Means and mean square errors of the pH values at 45 min and 24 h post-mortem measured at the LT and ST muscles in the four weight groups and in male and female groups.

	Slaughter Liveweight				Sex		*p*-Value			
	10 kg *n* = 14	15 kg *n* = 14	20 kg *n* = 14	25 kg *n* = 14	M *n* = 28	F *n* = 28	Weight	Sex	W × S	MSE
pH 45 *LT*	6.41	6.50	6.56	6.48	6.50	6.47	0.3778	0.6507	0.0705	0.05
pH 24 *LT*	5.72 ^a^	5.62 ^a^	5.91 ^b^	5.73 ^a^	5.76	5.73	0.0032	0.5562	0.1407	0.04
VarpH 45-24 *LT*	0.69 ^a^	0.88 ^b^	0.64 ^a^	0.75 ^a,b^	0.74	0.74	0.0111	0.9394	0.1036	0.04
pH 45 *ST*	6.14	5.95	6.23	6.01	6.06	6.10	0.1100	0.8447	0.0487	0.08
pH 24 *ST*	5.76 ^a^	5.72 ^a^	6.00 ^b^	5.76 ^a^	5.82	5.80	0.0141	0.7363	0.5515	0.06
VarpH 45-24 *ST*	0.38	0.23	0.23	0.25	0.24	0.30	0.5601	0.5280	0.1304	0.10

MSE = mean square error. M: male; F: female. VarpH 45-24: pH drop from 45 min to 24 h post-mortem. W × S: Weight **×** Sex interaction. ^a,b,^: Means in the same row with different letters are significantly different (*p* ≤ 0.05).LT: Longissimus thoracis muscle; ST: Semitendinosous muscle.

**Table 2 animals-11-01323-t002:** Means and mean square errors of daily colorimetric parameters of LT muscle in the four weight groups and in male and female groups during the five days post-mortem.

	Slaughter Liveweight				Sex		*p*-Value			
	10 kg *n* = 14	15 kg *n* = 14	20 kg *n* = 14	25 kg *n* = 14	M *n* = 28	F *n* = 28	Weight	Sex	W × S	MSE
L*1	43.39 ^a^	41.86 ^a,b1^	40.24 ^b,c^	38.84 ^c^	41.36	40.80	0.0000	0.3534	0.2415	5.07
L*2	45.91 ^a^	44.08 ^b2^	42.16 ^c^	40.90 ^c^	43.57	42.95	0.0000	0.2958	0.6295	3.89
L*3	44.58 ^a^	43.15 ^a12^	41.22 ^b^	39.40 ^b^	42.40	41.77	0.0000	0.3375	0.1276	3.95
L*4	44.09 ^a^	42.84 ^a12^	41.01 ^b^	40.07 ^b^	42.37	41.64	0.0002	0.2243	0.1832	3.83
L*5	44.04 ^a^	42.63 ^a,b1^	41.13 ^b,c^	40.08 ^c^	42.38	41.56	0.0006	0.1862	0.1703	3.95
*p*-Value	0.1610	0.0025	0.4084	0.0561	0.1023	0.3913	Ageing effect			
a*1	12.81 ^a^	13.84 ^b^	15.45 ^c1^	15.67 ^c1^	14.37 ^1^	14.52	0.0000	0.5755	0.0555	1.02
a*2	11.59 ^a^	12.48 ^a^	14.09 ^b2^	14.16 ^b2^	13.12 ^2^	13.04	0.0000	0.8140	0.4126	1.39
a*3	12.41 ^a^	12.69 ^a^	14.25 ^b2^	14.22 ^b2^	13.23 ^2^	13.55	0.0005	0.3823	0.0380	1.18
a*4	12.55	12.92	14.07^2^	13.88 ^2^	13.05 ^2^	13.67	0.0562	0.1645	0.8182	2.08
a*5	12.21	12.63	13.24 ^2^	13.72 ^2^	12.52 ^2^	13.38	0.1396	0.0690	0.5847	2.26
*p*-Value	0.4010	0.0275	0.0026	0.0072	0.0022	0.2754	Ageing effect			
b*1	3.95 ^1^	4.01 ^1^	4.54 ^1^	4.54 ^1^	4.27 ^1^	4.24 ^1^	0.1181	0.9010	0.2829	0.71
b*2	4.92 ^a2^	6.17 ^b2^	6.25 ^b2^	6.10 ^b2^	5.94 ^12^	5.77 ^2^	0.0128	0.5422	0.8230	0.83
b*3	5.92^23^	6.58^2^	6.88 ^2^	6.70 ^2^	6.72 ^12^	6.32 ^2^	0.4982	0.3571	0.4280	1.78
b*4	5.91 ^a23^	6.76 ^a b2^	7.32 ^b2^	6.79 ^a b2^	6.88 ^12^	6.52 ^2^	0.1136	0.3432	0.4188	1.54
b*5	6.05 ^3^	6.89 ^2^	7.29 ^2^	7.28 ^2^	7.06 ^2^	6.70 ^2^	0.0817	0.2986	0.4339	1.23
*p*-Value	0.0003	0.0000	0.0001	0.0001	0.00001	0.0002	Ageing effect			
C*1	13.42 ^a^	14.43 ^b^	16.12 ^c^	16.34 ^c^	15.02	15.14	0.0000	0.6418	0.0979	1.03
C*2	12.61 ^a^	13.95 ^b^	15.43 ^c^	15.46 ^c^	14.43	14.29	0.0000	0.6858	0.4166	1.31
C*3	13.77 ^a^	14.35 ^a^	15.90 ^b^	15.83 ^b^	14.91	15.02	0.0000	0.7093	0.1601	0.85
C*4	13.91 ^a^	14.65 ^a^	15.92 ^b^	15.60 ^b^	14.82	15.22	0.0003	0.2119	0.9585	1.07
C*5	13.65 ^a^	14.47 ^a b^	15.16 ^b^	15.61 ^b^	14.44	15.00	0.0092	0.1518	0.5622	1.56
*p*-Value	0.4106	0.2326	0.2594	04123	0.7772	0.9592	Ageing effect			
H*1	17.13 ^1^	16.25 ^1^	16.32 ^1^	16.20 ^1^	16.54 ^1^	16.41 ^1^	0.8466	0.8819	0.1220	10.14
H*2	23.12 ^2^	26.31 ^2^	24.04 ^2^	23.31 ^2^	24.42 ^2^	23.98 ^2^	0.1423	0.6891	0.7247	13.47
H*3	25.54 ^2^	27.43 ^2^	25.93 ^23^	25.27 ^2^	27.06 ^23^	25.03 ^2^	0.7941	0.2620	0.1467	29.52
H*4	25.30 ^2^	27.85 ^2^	27.65 ^23^	26.35 ^2^	27.96 ^23^	25.62 ^2^	0.7627	0.2070	0.4847	36.13
H*5	26.25 ^2^	28.93 ^2^	29.01 ^3^	28.21 ^2^	29.54 ^3^	26.67 ^2^	0.6843	0.0884	0.4829	28.75
*p*-Value	0.0004	0.0001	0.0001	0.0002	0.0003	0.0004	Ageing effect			

MSE = mean square error. M: male; F: female. ^a,b,c,^: Means in the same row with different letters are significantly different (*p* ≤ 0.05). Ageing effect. Means in the same column with different numbers are significantly different (*p* ≤ 0.05). *n* = 9 (three shots per determination and three measurements per sample were taken to obtain a single mean color value). W × S: Weight **×** sex interaction.

**Table 3 animals-11-01323-t003:** Means and mean square errors of moisture, WHC and cooking loss of LT muscle in the four weight groups and in male and female groups.

	Slaughter Liveweight				Sex		*p*-Value			
	10 kg *n* = 14	15 kg *n* = 14	20 kg *n* = 14	25 kg *n* = 14	M *n* = 28	F *n* = 28	Weight	Sex	W × S	MSE
% Moisture	77.77	77.77	76.48	75.48	77.04	76.70	0.0561	0.6188	0.5125	6.40
WHC	11.52 ^a^	13.15 ^a^	15.39 ^b^	17.15 ^b^	14.99	13.62	0.0000	0.0583	0.7743	7.04
CL	1.82 ^a^	3.62 ^b^	2.40 ^a^	1.74 ^a^	2.26	2.53	0.0093	0.5147	0.9723	2.47

WHC: Water holding capacity (expressed as percentage of water expelled); CL: Cooking loss (expressed as a percentage). MSE = mean square error. M: male; F: female. W × S: Weight **×** Sex interaction. ^a,b^: Means in the same row with different letters are significantly different (*p* ≤ 0.05). *n* = 3 three replicates were studied for each biological sample.

**Table 4 animals-11-01323-t004:** Means and mean square errors of the percentage of IMF samples obtained from the QFQ, *BF, SM* and *SE* muscles, in the four weight groups and in male and female groups.

	Slaughter Liveweight				Sex		*p*-Value			
	10 kg *n* = 14	15 kg *n* = 14	20 kg *n* = 14	25 kg *n* = 14	M *n* = 28	F *n* = 28	Weight	Sex	W × S	MSE
IMF-*QF*	2.29	2.42	3.02	3.01	2.46	2.92	0.0596	0.0270	0.8129	0.57
IMF-*BF*	2.15 ^a^	2.31 ^a^	2.97 ^b^	3.24 ^b^	2.41	2.92	0.0009	0.0166	0.5069	0.58
IMF-*SM*	2.97	2.86	3.92	3.75	3.12	3.63	0.2182	0.2460	0.2812	1.05
IMF-*SE*	2.39 ^a^	2.61 ^a^	3.49 ^b^	3.61 ^b^	2.87	3.18	0.0000	0.1081	0.6403	0.47

IMF: Percentage of intramuscular fat; IMF-QF: percentage of intramuscular fat in the quadriceps femoris muscle; IMF-BF: percentage of intramuscular fat in the biceps femoris muscle; IMF-SM: percentage of intramuscular fat in the semimembranosus muscle; IMF-SE: percentage of intramuscular fat in the supraespinatus muscle. MSE = mean square error. M: male; F: female. W × S: Weight **×** Sex interaction. ^a,b,c^: Means in the same row with different letters are significantly different (*p* ≤ 0.05). *n* = 3 three replicates were studied for each biological sample.

**Table 5 animals-11-01323-t005:** Means and mean square errors of the percentage of total collagen, insoluble collagen and soluble collagen extracted from the QF, BF and IE muscles in the four weight groups and in male and female groups.

	Slaughter Liveweight				Sex		*p*-Value			
	10 kg *n* = 14	15 kg *n* = 14	20 kg *n* = 14	25 kg *n* = 14	M *n* = 28	F *n* = 28	Weight	Sex	W × S	MSE
*QF* -%TC	0.68 ^a^	0.62 ^a b^	0.52 ^b^	0.58 ^a,b^	0.66	0.54	0.0055	0.0005	0.8388	0.01
*BF*- %TC	0.86	0.73	0.78	0.80	0.83	0.76	0.2695	0.1054	0.4930	0.03
*IE*- %TC	1.03 ^a^	0.79 ^b^	0.64 ^b^	0.65 ^b^	0.74	0.81	0.0070	0.4664	0.9016	0.10
*QF*- %IC	0.50	0.42	0.40	0.42	0.46	0.41	0.0521	0.1183	0.3235	0.01
*BF*- %IC	0.57 ^a,b^	0.53 ^a^	0.57 ^a,b^	0.67 ^b^	0.63	0.54	0.0405	0.0108	0.6808	0.02
*IE*- %IC	0.51	0.42	0.46	0.45	0.48	0.44	0.1175	0.1732	0.4516	0.01
*QF*- SC	0.27 ^a^	0.27 ^a,b^	0.18 ^b^	0.18 ^b^	0.24	0.21	0.0000	0.0998	0.4731	0.00
*BF*- SC	0.30 ^a^	0.28 ^a^	0.22 ^b^	0.18 ^c^	0.25	0.24	0.0000	0.6544	0.4735	0.00
*IE*- SC	0.36 ^a^	0.31 ^a^	0.23 ^b^	0.21 ^b^	0.28	0.28	0.0000	0.9622	0.8827	0.00

TC: total collagen; IC: insoluble collagen; SC: soluble collagen. *QF: quadriceps femoris*; *BF: biceps femoris*; *IE: infraespinatus*. MSE = mean square error. M: male; F: female. W × S: Weight **×** Sex interaction. ^a,b,c,^: Means in the same row with different letters are significantly different (*p* ≤ 0.05). *n* = 3 three replicates were studied for each biological sample.

**Table 6 animals-11-01323-t006:** Means and mean square errors of the instrumental texture parameters in raw meat and in cooked meat in the four weight groups and in male and female groups.

	Slaughter Liveweight				Sex		*p*-Value			
**Raw Meat**										
	**10 kg** ***n* = 14**	**15 kg** ***n* = 14**	**20 kg** ***n* = 14**	**25 kg** ***n* = 14**	**M** ***n* = 28**	**F** ***n* = 28**	**Weight**	**Sex**	**W × S**	**MSE**
WB										
Shear force (N)	20.54 ^a^	22.81 ^a b^	25.69 ^b^	25.70 ^b^	26.12	21.25	0.0230	0.0007	0.0244	2563.54
TPA										
Hardness (N)	17.49	18.68	19.08	18.46	19.00	17.86	0.7863	0.3204	0.0659	1853.00
Springiness (N)	3.97	4.30	4.48	4.28	4.36	4.14	0.6055	0.4114	0.1746	105.59
Chewiness (N)	2,312,682.66	2,611,216.70	2,875,192.10	2,670,409.63	2,796,238.76	2,438,521.59	0.6841	0.2820	0.0904	23,913,727.74
**Cooked Meat**										
	**10 kg** ***n* = 14**	**15 kg** ***n* = 14**	**20 kg** ***n* = 14**	**25 kg** ***n* = 14**	**M** ***n* = 28**	**F** ***n* = 28**	**Weight**	**Sex**	**W × S**	**MSE**
WB										
WBSF (N)	18.67	15.66	19.58	18.64	18.54	17.74	0.4992	0.6750	0.8403	5195.63
TPA										
Hardness (N)	3330 ^a^	3661 ^a,b^	3732 ^a,b^	3931 ^b^	3584	3743	0.0349	0.2666	0.8768	280,807.00
Springiness (N)	5.63	5.83	5.78	5.87	5.56	5.99	0.9306	0.1293	0.8232	110.26
Chewiness (N)	7,751,833.21	8,738,166.45	9,313,934.48	9,896,184.71	8,529,824.17	9,320,240.16	0.1783	0.2643	0.8512	6,990,082,054.00

MSE = mean square error. M: male; F: female. W × S: Weight **×** Sex interaction. ^a,b^: Means in the same row with different letters are significantly different (*p* ≤ 0.05). *n* = 6 six replicates were studied for each biological sample.

## Data Availability

Not applicable.

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
