# Peer review of "Liveweight and Sex Effects on Instrumental Meat Quality of Rubia de El Molar Autochthonous Ovine Breed"

_animals, 2021, doi:10.3390/ani11051323_

Round 1

Reviewer 1 Report

Question 1.  2.2. Sampling. In sheep meat quality studies, the longissimus dorsi muscle is the most used muscle, together with the semimembranosus, semitendinosus, biceps femoris and triceps brachii. What criteria were followed to select the muscles used in this study?

Question 2.- Line 136

after waiting one minute to let fresh air in contact with the meat….”

Don't you consider the time indicated for meat oxygenation to be too short, since the standardization of methodologies to evaluate meat quality recommends 1 hour of oxygenation (blooming)?

Question 3.- Line 205

Sex was not taken into account”

Since this is the first study to be carried out in this breed, would it not have been interesting to determine the effect of sex on aging time on colorimetric parameters?

Question 4.- Line 211

In line 125 and 126 it indicates that the ph measurements were taken at 0 hours (just after carcass dressing) and after a 24 hour-chill, but it does not indicate at 45 minutes and in the results (line 211 and table 1) it does not indicate the ph values at time 0 but at 45 minutes. Could you explain it?

Question 5.- Table 2.

After 3 days of ageing (a*3), the interaction W*S results significant, could you explain it?

Question 6.- Table 2.

Significance for H*3 at 10kg is not indicated.

Question 7.- Line 241-242.

“Slaughter weight only had a significant effect on b* at 48 hours post mortem (P≤0.05), increasing its value from 5.91 (10 kg) to 7.32 (20 Kg)”, but these values corresponded to b*4.

Question 8.- Line 246.

(P≤0.001) instead of (P≤0.01).

Question 9.- Line 251.

Table 2 instead of Table 3.

Question 10.- Line 289 and Table 3.

QF instead of Q.

Question 11.- Line 301.

In line 171 it is indicated that samples of QF, BF, SM and IE were collected for SC determination, but Table 5 does not include the results of IE.

Question 12.- Line 315.

“Sex had effect on the percentage of IC in BF”. Could include its values in the same way as it did when describing TC.

Question 13.- Table 6.

In WBSF in raw meat. the interaction W*S results significant, could you explain it?

Question 14.- Line 397.

It might be more appropriate to include "In general, b* parameter was no affected… ", because when going from 10 to 15 kg, the effect of the slaughter weight was observed.

Question 15.- Line 430.

it is mentioned that “no effect of sex on moisture, WHC or CL was detected “, but the differences of CL according to slaughter weight are not explained.

Question 16.- Line 455.

It is mentioned that “IC content was affected by slaughter liveweight for BF muscle” but the differences according to sex are not considered.

Question 17.- Line 509 and 513.

The references cited in lines 509 and 513 (meat sensory parameters assessed by trained panelists) could not be included since the work they referred to is still under review.

Question 18.- Line 517.

No conclusions on collagen content are included.

Author Response

Thank you very much for the work of reviewing the manuscript. Your comments have been of great help to improve it. We have tried to answer your questions and improve the background and form defects of the manuscript.

We answer your arguments in an attached word file.

Best regards

Reviewer 2 Report

Presented for review work gives information about instrumental quality of Rubia de El Molar meat, danger of disappearing autochtonous sheep breeds. The processes of displacing local indigenous breeds were common in many countries. Actions related to the protection of genetic resources allowed for the saving of many valuable breeds, with lower productivity, but extremely important for a natural management based on extensive pastures plays, as a component of landscape, folk culture and supplier of many valuable products. The lamb’s meat fully match the current needs of modern consumers looking for tasty and also healthy food.

In the work analyzed a number of different physical-chemical parameters. The authors concluded that differences observed in instrumental parameters could explain the differences observed in sensory parameters but generally weight had a greater effect than sex on Rubia de El Molar meat quality characteristics. Experimental design, lab methodology and statistical analysis is correct.

Comments: Research relating to on the influence of various factors on the quality of lamb meat are quite numerous so in my opinion the application aspect of the work is necessary, it is worth drawing conclusions that could indicate the most appropriate slaughter date.

Please unify the record of abbreviations and markings under the tables: the most readable is the following: explanation of symbols, then the significance of differences as in table 3 or 4; in table 5 there are no explanations.

Presented for review article can be publish in Animals after minor revision.

Author Response

(The authors gave the same response as above.)

Reviewer 3 Report

This manuscript by Miguel and colleagues investigated the effect of liveweight and sex effects on instrumental meat quality of Rubia de El Molar autochthonous ovine breed, a local specie in Spain. This is the first work studied the instrumental quality of Rubia de El Molar suckling lamb meat. This is an interesting topic. It can be accept after the following revision.

  1. Lines 3-4, two authors labeled * as Correspondence, however only one contacted information was labeled as *Correspondence: [email protected]; Tel.: +34-91-8879410 in the line 9.
  2. Abstract, authors need to simply described the experimental design. How many lamb used for each group? Age of the lamb? And so on.
  3. Abstract, authors need to add the detailed information about degree of the changes for each parameters.
  4. Line 31, “(measured by a Texture Profile Analysis test)” could write in the method section. Do not too much details in the abstract.
  5. Introduction, paragraph 1-2, no reference; and check others with the same problems.
  6. Introduction, many paragraph only have one or two sentence. Please combine them logically.
  7. Lines 60, 64, 72 and etc., “cues[1-4]”, “11 kg[1]” and “[7]have”; space needed between “[]”and words. Please check throughout the manuscript.
  8. Lines 108-116, please remove “” from the text. Many paragraphs only have one or two sentence. Please combine them logically. There are many similar problem throughout the manuscript, please correct all of them.
  9. Tables 1-6, please check the requirement of the journal style of Animals. It is very weird to use the current style. The authors need to respect the journal style and put effort to revise them.
  10. In all the tables, please use the P value to substitute ‘NS’.
  11. In all the tables, please correct ‘Sign.’ To ‘P value’ in the table.
  12. In all the tables, please added the replicates of each measurements as n=?.

Author Response

(The authors gave the same response as above.)

Round 2

Reviewer 3 Report

No further comments.

Author Response

Thank you very much for your help in reviewing the manuscript.
